# Practical Theology and Social Just Pedagogies as Decoloniality Space

## John Klaasen [1,2] 

1    Department of Religion and Theology, University of the Western Cape, Cape Town 7535, South Africa; jsklaasen@uwc.ac.za
2    Theology, Diaconia and Leadership Studies, VID Specialized Unievrsity, 0370 Oslo, Norway

**Abstract:** Higher education institutions in South Africa are still dominated by colonial traditions, course content, staff with colonial privileges and attachments, and discriminatory structures and systems. Practical theology and theologians are no exception. This article seeks to investigate the correlations between social just pedagogies and social justice. Social just pedagogies consider the role of the students, lecturers, and non-human phenomena as contributing to epistemology and agency formation. Normative pedagogies remain important criteria for knowledge production and graduate attributes within the South African higher education landscape. Within practical theology, the pedagogies that are used to form students and impart knowledge are still dominated by classical teaching methods that are power-centred and biased towards the privileged. The aim of this article is thus not to replace the normative pedagogies but to challenge the normativity and essentialism that has characterised colonial, race-related, and top-down knowledge production. I will introduce a social just pedagogy of teaching practical theology that critically engages and challenges the privileged normative position of classical practical theology. A social just pedagogy will bring the centre of learning and teaching into the structure of the lecture room, a participatory method of knowledge production, students, and the lecturers. The hierarchical structure of the South African university system will be engaged with as an instrument of traditional classical knowledge production systems. Teaching practical theology through social just pedagogies will also contribute to social justice within democratic South Africa. The question that I will address is how teaching practical theology at higher education institutions can contribute to the agency of social justice in South Africa.

**Keywords:** South African higher education; decoloniality; social justice; social just pedagogies; practical theology; learning; teaching

## 1. Introduction: Decoloniality, Social Just Pedagogy, and South Africa

Higher education institutions (HEIs) in South Africa are still dominated by colonial traditions, course content, staff with colonial privileges and attachments, and discriminatory structures and systems. Practical theology and theologians are no exception. Post-democratic South Africa has become a complex society, with the continued influence of colonial and imperial social and political orders dominating the higher education (HE) landscape. This complexity has manifested in the recent student protests of the last decade. Responses to the student protests and the continued symbolic and structural presence of colonialism and imperialism within a relatively young democracy such as South Africa can be summarised in the different and varied efforts of decolonisation. Whilst decolonial, decoloniality, and colonialism have challenged the dominant Western and classical approaches of developed and technological advanced countries and the underdeveloped and less technological countries, my own contribution is a critical investigation of the binary of knower and learner, or teacher and learner, within HEIs such as the University of the Western Cape (UWC), which is categorised as a former black HEI. I use the concept of

'decoloniality' to distinguish it from 'colonialism' and 'decolonial' because of the contestations of the terms. Decoloniality refers to an "invisible power structure that sustains colonial relations of exploitation and domination long after the end of direct colonialism" (Ndlovu-Gatsheni 2012, p. 48). As Sakupapa contends, decoloniality is "an analytical category to give expression to the continuity of coloniality and its manifestation in the academy in general and in theological education in particular" (Sakupapa 2018, p. 408).

This contribution seeks to investigate the correlations between social just pedagogies and social justice to form the agency of students and staff within practical theology towards social just pedagogies as a means of decoloniality. Social just pedagogies consider the role of the students, lecturers, and non-human phenomena as integral contributors to epistemology and agency formation. Normative pedagogies, such as lecturer and student and subject and object categories of knowledge producers, remain important criteria for knowledge production and graduate attributes within the South African HE landscape. Within practical theology, the pedagogies that are used to form students and impart knowledge are still dominated by classical teaching methods that are power-centred and biased towards the privileged. It is not the intention to replace the normative pedagogies, but to challenge the normativity and essentialism that characterises colonial, race-related, giver and receiver, and top-down knowledge production. I will introduce a social just pedagogy of teaching practical theology that critically engages and challenges the privileged normative position of classical practical theology. A social just pedagogy will bring the structure of the lecture room into the centre of learning and teaching as an equal space; a participatory method of knowledge production of equality; students and the lecturers as co-producers of knowledge; and non-living objects, such as teaching material and educational organs. The hierarchical structure of the South African university system will be engaged with as an instrument of traditional classical knowledge production systems. Teaching practical theology through social just pedagogies will also contribute to social justice within democratic South Africa. The question that I will address is how teaching practical theology at HEIs can contribute to the agency of social justice in South Africa that is dominated by colonial history, symbols, and structures. These colonial histories, symbols, and structures are embedded in the educational system and content of HEIs.

Inequalities have remained a major concern in democratic South Africa, and it has reached new heights. At the UWC we are constantly reminded of the historical effects of unequal education and the persistent gap between the rich and the poor by our student population make-up, and the disparity of the level and qualifications of the academic staff. As a former historically black university, the students are mainly from the poorer parts of South Africa and there is a growing student population from other parts of Africa, including refugees and migrants. "With the democratic dispensation post-1994 and the opening of South African society, increasing numbers of students and academics from the rest of Africa have entered the country. Despite official policies of welcome, this opening up has been met with outbreaks of xenophobia, in 2008 and 2015 (Aljazeera 2015; Human Rights Watch 2008)" (Leibowitz and Bozalek 2016, p. 111). While the top level of the academic staff are predominantly from privileged backgrounds, the lower level of staff—who are the majority—are from the marginalised, formerly oppressed section of the population (Leibowitz and Bozalek 2016, p. 109).

In a recent article, Clowes et al. (2017) reported their research findings of the third-year student group in Gender Studies at the UWC. They concluded that the narratives of the students:

> reinforces research showing how intellectual development and engagement at South African institutions of higher education is simultaneously a deeply social experience that is always already implicated in reproducing the hierarchies, exclusions and privileges characterizing contemporary society. The institution and all those who are part of it-are entangled in complex social and structural dynamics of unequal subjectivities, professional councils, higher education frameworks,

overarching neoliberal frames both nationally and globally that shape processes and possibilities . . . . (Clowes et al. 2017, p. 1)

Coupled with the historical inequalities and the effects on the infrastructure and resources of the UWC, COVID-19 has placed huge infrastructural and financial burdens on both the students and the institution of such universities. Some of these challenges are highlighted with the #FeesMustFall and #RhodesMustFall campaigns, the failures of the policies and legislature including The Department of Education's *White Paper of 1997*, The Soudien Report, The Department of Higher Education and Training's (DHET) paper of 2010, as well as recommendations from academics and activists (Clowes et al. 2017, p. 103). "Despite post-apartheid policy intentions to redress the effects of apartheid, inequalities in HE have remained an endemic problem in South Africa" (Bozalek and Zembylas 2017, p. 1).

Social just pedagogies target structural and institutional colonial practices. These kinds of pedagogies critique both the historical colonial and imperial systems and structures, and the use of such systems and structures by those who were previously powerless and are now part of traditional and classical colonial institutions and structures of society. This is the ongoing influence of past direct colonialism on the functions and processes within institutions. In more concrete terms, the lecturer–student relationship is challenged, and the human and non-human interaction is considered for a more interactionist movement of knowledge production. This approach examines the three purposes of education, namely: (i) is the purpose of education to complete a curriculum and obtain a qualification?; (ii) is it to socialise people as members of a community or citizens of society?; or (iii) is it for individuals to existentially respond to the problem of society? The first purpose refers to *qualification*, the second to *socialisation*, and the third to *subjectification* (Biesta 2009).

In view of the slow transformation in HEIs and the limitations of the normative pedagogies used at universities and colleges, social just pedagogies consider the students and non-human phenomena, including the lecture room space, and the virtual space, as more equal partners with the lecturers for effective knowledge production and a more sustainable just society. Social justice is a contested phenomenon that has been approached in various ways. I will use the notion of social justice within the broad framework of Martha Nussbaum (2002) and attempt to relate social justice to the agency of those at the margins.

## 2. Three Approaches

Kathleen Cahalan (2005) provides a typology of practical theology approaches that are responses to the modern project. Her approaches include a continuation of the modern project (the late moderns), a break from the modern project (countermoderns), and radical postmoderns. The late moderns seek to finish and continue the modern project. Jurgen Habermas and Charles Taylor, although critical of the absolute autonomous individual of Kant, put the subject within situations that are characterised by political and dialogical engagement. Reason, unlike Kant, is not the sole character or power that constitutes an individual, but reason's power is within a community of engaged individuals.

### 2.1. The Late Modern Approach

Don Browning is a representative of the late modern approach. He found conversation partners in Ricoeur, Gadamer, and Habermas. He revised reason as embedded in praxis and argued that the telos of practical theology is practical concerns and questions that the pluralistic age is posing to Christianity. Theology must be able to dialogue and critically engage with both religious and non-religious claims and practices (Cahalan 2005, p. 68). He was one of the first practical theologians to implement an interdisciplinary approach within practical theology. As far back as the mid-sixties when he was busy with his PhD, Browning used psychotherapy and theology in a correlational manner (Hestenes 2012, p. 3; Klaasen 2014, p. 2). Browning (1985) drew from both tradition and culture, from doctrines and lived experience, and from reason and non-reason traditions.

Browning does not discard universally situated reason, although experience and knowledge are culturally and historically embedded. There are some arguments that are

better or worse than others that determine what is right or what is wrong. "This formal, structural constant is expressed in the human capacity for reversable thinking, the basis for understanding our obligations of equal regard, mutuality and agape". Principles such as 'love for neighbour' and the 'golden rule' are found in both religious and cultural practices (Cahalan 2005, p. 69).

Browning concludes that education must move beyond the clerical paradigm so that ministers engage the congregation for the transformation of both the religious and the larger society. Ministers must be trained to lead congregations in practical reason in order to engage both the religious and the secular perspectives (Cahalan 2005, p. 72). "Practical theology's task includes ministry to the church and the world. In another clear breakaway from the theory–practice dichotomy, Browning developed Farley's idea of practical theology beyond clericalism" (Klaasen 2014, p. 2). "Both the inner-ecclesial and public foci of these activities would be a part of the concerns of practical theology" (Browning 1985, p. 16; 1991, p. 57).

There are some definite points of contact between Browning's modern practical theology approach and social just pedagogy, although Browning did not refer specifically to knowledge production. Like social just pedagogy, the aim of Browning's approach is both clerical and social, the church and the world. Put differently, knowledge is not just abstract or only for the activities and mission of the church. Knowledge is for the purpose of challenging structural power relations and the relationship between theory and concrete situations. Ultimately, knowledge is for the purpose of a qualification.

### 2.2. The Countermodern Approach

The second approach opposes the modern project in favour of a confessional one that derives from Christian practices and beliefs. Unlike Browning's politically and socially engaged practice, this approach seeks to strengthen Christian practices and identity as normative for engagement. Alistair MacIntyre (1981) is regarded as the one who established this approach through his highly acclaimed book *After Virtue.* According to McIntyre, virtues are cultivated through practices over time within groups of persons. Identity and character are formed within specific traditions which the community accepts as valuable for virtuous and moral persons. The Christian ethicist Stanley Hauerwas was another proponent of the communitarian moral formation perspective, in opposition to the modern project's individualistic perspective on identity and moral formation.

Practical theologians Dykstra and Bass (2002) have identified twelve practices that, if integrated together, form a way of life and address the underlying human needs. The practices are "honouring the body, hospitality, household economics, saying yes and saying no, keeping Sabbath, testimony, discernment, shaping communities, forgiveness, healing, dying well, singing our lives" (Dykstra and Bass 2002, p. 16). These practices, when coordinated in a continuing process, form a way of life that is in line with the Christian tradition and is passed on by static doctrines and confessions. These narratives are part of a metanarrative that does not need to be defended against rivals or opposition. Tradition, scripture, and the narratives of biblical figures and historical Christian events are enough to maintain traditional norms and values and counter the modern project of individualism, abstract reason, and universal principles.

Modern patterns brought about by technology in favour of traditional ways of life, rhythms, and habits that are altered by universalism, and the rise of secularism and decline of religion, can be resisted by a way of life that is transmitted through traditional wisdom and practices. Instead of a well-defined methodology like Browning's, countermoderns rely on certain claims and doctrines about who we are and how the Christian narrative provides a basis for such doctrines and claims.

The Christian story as a metanarrative is enough for educational formation, and the tradition, community, and doctrines of the church are sufficient to address all the challenges of modernity. Education is doctrinal, traditional, and clerical. The minister is trained to

defend the traditions and teach it to the congregation. Minsters are trained, modelled, and mentored into countering the values, life patterns, and practices of modernity.

The purpose of countermoderns is to socialise people in a particular tradition with certain habits, traits, virtues, beliefs, and behaviours. Tradition and community have a greater role than reason, and the wisdom of the historical figures and events take the place of universal and abstract reason. Whilst this approach is closely related with the formation of agency of socially just pedagogies, reason plays a major role in the formation of agents for social justice. It is not so much historical and traditional wisdom that emerges in interaction and relationships, but new engaged reasoning.

### 2.3. Radical Postmodern Approach

This approach is represented by philosophers such as Derrida, Foucault, and French feminism. Within theology there is a broad spectrum that includes liberation theology, contextual theology, Black theology, and Feminist theology, who take the local culture and experience seriously as a lens for practicing theology and interpreting lived realities.

This approach combines reason and actions as praxis and makes little distinction. It also challenges universalism in favour of relativism and rejects the normativity of European, male theologies. Denise Ackermann (2008) is one of the foremost practical theologians who adopted this approach as one that confronts the marginalisation of persons and the disembodiment of the poor and suffering. Confronting the dominant use of white male experience concerning theology, Ackermann called for a human experience as inductive. Within Black theology, Tshaka, Maluleka, and Boesak can also be mentioned.

This approach raises questions of power relations within educational systems and approaches. It puts the marginalised and oppressed within the centre of education. The experiences are not limited to the church, but the world is as part of the experience as that of the faithful. The training of ministers is geared towards critical action of the situations of congregations.

The purpose is subjectification, and education plays a role in the conscientisation of students to solve the problems of society. This is a bottom-up movement, and culture, context, and lived experience are preferred over doctrine and tradition. The failure to be critical of context makes culture, experience, and relativity absolute. The poor, marginalised, privileged, and oppressed are placed in the centre and in positions of power that can cause harmful power relations, as well as uncritical and selfish binary identities.

My early approach was within the late modern one that engages with reason and practice in a dialogical way. I adopted Browning's critical correlation approach, which is interdisciplinary and practice-orientated. Browning's five levels, and later five dimensions of ethics, is an attempt to combine reason with experience, and it is the comparative status of experience that makes this approach both relevant and applicable within postmodernity. My own critique has been his distance between reason and experience. "Browning still seems to give a privileged position to rationality. By doing this theory absorbs practice to the point where practice loses its value in the hermeneutical process" (Klaasen 2014, p. 3). Practice is the primary level in the process. In practical terms, ecclesial practice like preaching, education, pastoral care, worship, and ministry is the basis for practical theological education. Through reflective engagement, reason interacts with experience in a dialectical manner so that experience maintains independence. There is a greater distance between reason and experience than in Browning's critical correlation approach.

My later approach leaned towards the radical postmodern approach with an emphasis on narrative and the interpretation of cultural, social, and political experiences as both an interpretative tool and substantive assertion. Narrative changes power relations and places the experiences of the marginalised at the centre. The students become as important as the lecturers within knowledge production.

Rosemary Radford Ruether's (1983) dialectical approach to theological education challenges ministry as a symbiotic process that does not constitute dualisms or either/or but gives rise to new forms of inclusive ministry. The marginalised groups, such as those

discriminated against on the basis of race, gender, sexuality, disability and order, are put in creative tension with the powerful, the clerical, the normative, and traditionalists.

These three approaches represent a significant corpus of practical theologians. The three approaches make valuable contributions to the formation and education of ministers (both lay and ordained). The critical engagement of reason, universalism, and individualism of modernity gives rise to innovative and new approaches to the educational structures, paradigms, and pedagogies within practical theology.

Within my own approach to practical theological teaching, I have used elements to a greater or lesser extent from each approach. However, in the later part of my theology teaching career, I have asked new questions that relate to the South African democratic society and the agency of students of practical theology.

### 3. Socially Just Pedagogy

The South African political and social landscape has come under renewed criticism because of the ongoing corruption, xenophobia, racism, poverty, and the growing gap between the rich and the poor. Practical theology as a scholarly discipline is concerned with the practical and theoretical engagement of actual experience and looks at innovative ways to address societal and religious matters. While the above three approaches of practical theology form the dominant landscape of the South African theological scene, I would like to present a socially just pedagogy for social justice as a more effective and comprehensive one.

Leibowitz and Bozalek contend that social just teaching and learning is reciprocal between the teacher and the student, and that it has a social justice orientation (Leibowitz and Bozalek 2016, p. 111). Social just pedagogy is centred around the principle of relationality or reciprocity. The process of knowledge or teaching and learning takes an interaction between different entities. In this case it is between the teacher or the learner and the curricula, learning processes, kinds of research, and advocacy (Leibowitz and Bozalek 2016, p. 112).

Social just pedagogy also assumes that equality and participation are perquisites. Participation is not restricted to the teacher or lecturer as is normative in classical teaching pedagogies. The equal participation of students adds to the degree of equality and the subsequent fostering of social justice. This entails social arrangements, both kinds of relationships between lecturer and student, and principles that govern processes, resources, and structures, which foster and enable equal participation. A sense of belonging necessitates a form of dialogical community and way of being that critically engages with socially constructed exclusions such as gender, race, or class. Instead, common good, common principles, common values, and common "political symbolism" should be the foundation of a deeper dialogical reflection than mere form of communication. "Dialogue should be part of a component of engagements that have reflexive self-problematization (reflexive justice) as [a] central goal. This may mean that academics and students should reflect not only on their own histories of marginalization, but also on histories of current and on-going privilege within and outside the frame of HE contexts" (Bozalek and Carolissen 2012, p. 13).

Social justice has the dual aim of ensuring an equal space for equal participation within the relationship of the subjects, and the vocation to actively commit to justice beyond their own spaces of HE. HEIs and the activity of teaching are catalysts for participation in social justice beyond the formal academic spaces. The barriers of the institution are symbolically crossed by a social just pedagogy that uses resources, policies, structures, and processes that are based on social justice principles, policies, and practices.

Conte-Frazier, when referring to the goal of practical theology to heal the world, claimed that religious education addresses this need when it teaches social justice. Paulo Freire (1970, p. 178) referred to the freedom that is embedded in the pedagogy of the action of the people. The fundamental point in these kinds of pedagogies is the extent of participation by the oppressed, the marginalised, and those at the periphery. In the context of HE, this refers to the freedom of students, particularly those that are black, female,

poor, and pilgrims. "Religious education for social justice is where the history and present realities of oppression no longer paralyze us but instead their hold on us is slowly and steadily overthrown. It is where the participants are no longer passive receptacles but full subjects, actors, and catalysts of a historical moment for change. This change begins not in the esoteric confines of the abstract but with everyday life, in the realm of *lo cotidiano*" (Conde-Frazier 2006, p. 321).

Bozalek and Carolissen (2012, p. 9) argued that "traditional normative frameworks" construct hegemonic educational discourses that bestow power on a selected category of knowledge producers. These discourses are usually associated with language and vocabularies. Within the South African contexts, language and vocabularies are closely aligned with race and the position of persons involved in the process of teaching and learning. It is therefore imperative to address the history of HE from both an internal and external history. Richard Niebuhr (1941) used 'internal' and 'external' as a continuum of the church's past as memory and the history of the world as the space to which the church is called to respond to in ministry and mission. When applying Niebuhr's notion of history as internal and external to education, and practical theology in particular, education at HEIs engages both the internal history of the institution and the staff and students, as well as the broader societal history.

Hegemonic discourses are prevalent in both traditionally white institutions of HE, and in institutions such as the UWC through the classical separation of lecturer and student positionality and the neglect of the non-living agency of the lecture space. While the recent research in decoloniality and social just pedagogies contributed to the lecturer and student power relations within education, the non-living space has not been given the same attention. Within practical theology, decoloniality is less researched, and social just pedagogies are almost non-existent, except in pedagogies such as participatory action research (PAR), and in the last forty years, the narrative approach. PAR encompasses a number of pedagogies and traditions that embody theory and practice. The partners within this type of research usually include the researcher and persons such as practitioners, community members, volunteers, and officials of the institutions that related to the research. Several practical theologians have given less excluded references to the partners of the researcher. Muller refers to these partners as "co-researchers" and, in my own research, I have continuously used this reference in an attempt to acknowledge the skills and epistemologies of those who would not be conventionally referred to as knowledge producers. " . . . persons are not restricted to the academic scholars or professional practical theologians, but also includes the persons who tell their stories. Those who tell their stories are referred to as 'co-researchers' because they have as much authority as the professional theologian in both the subject and method of research" (Klaasen 2017, p. 466).

The aim of social just pedagogies is not disembodied knowledge or knowledge for the sole purpose of a qualification. Qualification denotes those aspects that are necessary for preparing a person with the skills and knowledge for a professional life. The socialisation and subjectification in Biesta's second and third purposes of education falls within the scope of social just pedagogy. Socialisation refers to the habits and character to be part of a community. Subjectification refers to the processes of unique individuals who, through education, may be existentially challenged to respond to the world around them (Biesta 2009). The combination of the second and third purposes implies that education and knowledge goes beyond the individual (lecturer) and includes the students, non-living phenomena like the lecture room space, and the communities that encounter the problems that need to be addressed.

*An Example of Teaching within a Socially Just Pedagogy*

In the first semester of 2022, I taught a pastoral care module to final year Bachelor of Theology students. We adopted a socially just pedagogy, and the following were important principles that both the students and I agreed upon:

- We agreed on the pedagogy, although it was a completely new approach;

- We determined the course material, method, and timeframe of assessment, aims and objectives;
- The method of teaching would include both students and lecturers;
- Sharing of histories so that we understand our contexts;
- Sharing of views and perceptions within an environment of trust, respect, and openness.

At the end the module, the following evaluations were received from the students:

**Student 1:** *"My idea of social justice pedagogy is that through our learning I've developed lenses that recognizes what is the social issues within our structures that are unjust. Not only have my perception and perspective change[d], but specifically to recognise when injustices are happening in relation to what I am learning. The people that urgently need voices that advocate change is right in my very own community. This type of pedagogy is application orientated as well as knowledge that the study of religion cannot be unwoven from political and civil life."*

**Student 2:** *"I especially found all the resources very helpful as it focuses on different challenges in society and offers practical theology to overcome such challenges. The pedagogy of storytelling, narrative building has granted me deeper insight into why it is so important. When an individual is being given a chance to create narrative around and about their life, they develop agentic status."*

Pastoral counselling partners with sociology and anthropology in uncovering individual stories which, I have learned, is a powerful tool in pastoral care. Listening to how people construct their own story as a living document, and what grew out of them within that personal narrative, can bring hope, and, possibly, another or a new perspective.

**Student 3:** *"My expectation was that we would be given theory of pastoral care and do's and don'ts, but it was very different in a good way. It was also very relevant in the sense that it deals with issues directly from the perspective of people's personal experiences. The topics were also relevant and still valid in our communities and churches. It made me feel included in decisions which gave a sense of empowerment and gave room for freedom.*

*I was enlightened by the topics we dealt with, and it was interesting, since I have lived experiences such as my disabled son and could really reflect on how blessed and how positive our journey actually was. I enjoyed the Google Meets discussions."*

Three points can be highlighted from the evaluations that were received. Firstly, that the change of roles and the positioning of the students was not an easy transition from receiver of knowledge to equal participant in knowledge production. Secondly, the challenge towards agency and not simply passive recipient was a welcoming experience. Students reflected on the course material from a perspective of embedded knowledge and not just abstract reasoning. Thirdly, there was a greater awareness of injustice and the vocation to address issues and structures of injustice.

The introduction of storytelling within the social just pedagogy resulted in a different power relation between myself as a lecturer and the students. Their stories gave them a sense of power to paste together their episodes and formulate the plot that they wish to tell. It soon became apparent that there is a progression of roles and functions, and that both the lecturer and students were moving towards growth and wholeness.

## 4. Some Concluding Remarks

Firstly, the aim of socially just pedagogies is embedded in social just principles that foster the agency of social justice. The method of teaching and the content comprise social just principles which influence the positioning of each aspect of the module. The aim is to foster the participants, students, and lecturers into agents for transformation, as well as equality, fairness, and the basic needs to live a life of worth. This manner of addressing decoloniality challenges both the external structures and processes of inequality through agency formation of the students, and internally through the repositioning of power relations of the students, lecturers, and the physical space of the classroom.

Having the power to choose contributes towards a life of dignity. A life of dignity is understood within the framework of Nussbaum's political freedom that goes beyond fairness and embeds theory in practice. Nussbaum (2002) included definite and concrete social, political, and spiritual aspects in her notion of dignity. Bozalek and Zembylas (2017) used the feminist new materialist approach to demonstrate that matter is not restricted to what can be observed and touched, but the observed and concrete is intersected. Here, matter includes taking responsibility of the care of others, of activism for social justice and communal becoming, and participation towards a more just and equal society (Bozalek and Zembylas 2017, p. 76).

Secondly, social just pedagogies move beyond abstract and technical reason to socially embedded reasoning. This kind of reasoning is similar to the transversal reasoning of Muller and van Huyssteen. Socially-embedded reasoning happens within the lived experience of the marginalised, the poor, and discriminated. The collective experience of all participants within the pedagogical space supersedes the individual experience.

By lived experiences I refer to the diverse, religious, and non-religious, including spiritual, social, political, and cultural encounters and histories. Here, special attention is given to the telling of personal and communal stories as to how they are formed as part of the curriculum. The narratives of the students are incorporated in the content as part of assessments and evaluations, as well as objectives and aims.

Browning's ground-breaking post-foundationalism and the embeddedness of reason and Klaasen's theory, practice, theory, and practice approach is a helpful addition to Browning, Muller, and Van Huyssteen's ways of using reason within decoloniality. The contexts and experiences of students and their communities are interlinked with reflection on the course material. "Spaces of centres provide opportunities for students to bring their stories, experiences and perceptions, as critical epistemologies that engage normative, yet sometimes uncritical knowledge of the all-knowing lecturers. The teaching and learning space are not fixed but dynamic. The student, lecturer, and non-human phenomena rotate positions within these spaces so that learning and teaching become a process of growth, formation and enrichment" (Klaasen 2020, p. 6).

Thirdly, a reciprocal relationship exists between student, teacher, and non-human phenomena. I add the virtual space to the three phenomena. This aspect negates the longstanding tradition of the professor as the only expert of knowledge. The interchange of expertise, the openness to different kinds of knowledge, and the maximising of knowledge production is underlined by the reordering and repositioning of the classroom space. The classroom setting in a circle is symbolic of the interlinking and continuation of the different agents in the process of knowledge production.

Fourthly, students are important agents of both lived religious and non-religious experiences and form the basis for a hermeneutical lens that is applied in a multi-disciplinary context. Whilst students and staff may not have the knowledge of other disciplines, the openness and methodology used is inviting for proven scientific knowledge beyond the scope of practical theology. Agency here refers to the intellectual and moral formation of students for the fostering of social justice (Klaasen 2020, p. 4).

Fifthly, persons are trained within a socially just space with socially just principles to engage with the congregations through preaching, pastoral care, worship, and teaching in order to confront social injustices.

New models of ministry have been designed to address the imbalances and disparities both within the church and in society. More egalitarian models of ministry that challenge sexist, clerical, hierarchical, and domination forms of ministries have been added to approaches such as PAR and narrative approaches.

To conclude, religious education is both practical and theoretical—theoretical in a sense of the prescribed material that is developed over the period of the module to include the input of students, and practical insofar as it connects the rearranged classroom space with the virtual space and the localities of the students.

**Funding:** This research received no external funding.

**Institutional Review Board Statement:** Not applicable for the study did not involve humans.

**Informed Consent Statement:** Not applicable.

**Data Availability Statement:** Not applicable.

**Conflicts of Interest:** The author declares no conflict of interest.

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
