# Peer review of "Practical Theology and Social Just Pedagogies as Decoloniality Space"

_religions, doi:10.3390/rel14050675_

Round 1

Reviewer 1 Report

On a formal level, the text is well structured and up to standard, even though the clarity and objectivity of its well-written introduction is not kept up throughout. Particularly problematic appears to me section 2 (Three approaches, lines 132-271). For, after having set the stage, in the introduction, for engaging with the relationship between social justice and “the agency of those at the margins in South Africa” (line 70 and even more explicitly 130), the author(s) start with a theoretical discussion on different responses given to the project of modernity. Even the theorist considered to be able to furnish the most appropriate framework to reflect on this correlation, namely Martha Nussbaum (130), is only mentioned again in the concluding remarks (433), but rather en passant. Even though there undeniably is a connection between these three different responses to modernity and the issues of (post)coloniality and social justice, having omitted this discussion would have had no significant negative consequences on the paper’s overall outcome.

As far as the content is concerned, the article makes a valuable contribution to both the praxeological and theoretical scholarly discussion on (post)coloniality, social justice, and social just pedagogies. Even though some readers might be surprised by the paper’s method whereby the author(s) analyze their own practices themselves and normatively suggest it as a more appropriate normativity than others for their context, both the practice introduced at the University of the Western Cape and the theoretical reflection that arose from it add significantly to the academic debate.

At the same time, the theoretical framework of the proposed approach seems to me to be flawed on to counts: (a) no critical reflection on the normativity of one’s own approach—in this case, the social just pedagogies—is offered; (b) no the theological foundation of the envisioned Practical Theology informed by social just pedagogies is laid.

To be sure, the author(s) do not intend “to replace the normative pedagogies, but to challenge normativity and essentialism that characterises colonial, race related, giver and receiver, and top-down knowledge production (58-60)”. The question, however, arises whether two different, conflicting normative systems can at all be in place at once. Second, it is questionable whether any established normative system can be volitively replaced or abolished. In other words, what is effectively necessary to be able to have a new normative system established? Lastly, just as any other approach, social just pedagogies are by no means devoid of normativity. However, the latter is neither warranted nor made transparent in the article. Does this approach have any flaws, theoretical and praxeological?

As for (b), any innovative approach in any theological discipline requires a solid theological foundation, which is not provided with in this article. The justification and even legitimation of a theological approach cannot, in my view, be dependent of external rationales. To be recognized as theological, it should emerge from the very theological sources of revelation.

The positive aspect of this article is that it takes account of a series of new elements and factors and, most importantly, that it sets something new in motion, which has an impact on reality as well as on the lives of people involved in this process—as one can hear from the testimonies of the cited students.

The quality of English language is good. Please, just consider revising these two possible spelling mistakes:

- Line 138: “places”

- Line 336: “the no living space”

Author Response

The comments of the reviewer is very valuable and constructive. I appreciate that the reviewer recognized the contribution that this article brings to the reality of the lives of students and pedagogies used within South African universities. 

The theological foundation could have been more explicit and more specifically how Practical Theology can be informed by social just pedagogies. This particular contribution was aimed at laying a foundation for further research within the teaching of theology and Practical Theology in particular. The reviewers comments in this regard will be taken in consideration.

The pedagogy is not meant to be taken uncritically. However the approach engages critically with other normative approaches and within the contexts of the students and institutions where this research has been done. This approach provides critical engagement with traditional power relations and knowledge production. It highlights the role of students and the non living spaces that produces knowledge that is non conventional.

The comment with regard to the  structure of the text is appreciated and the reviewer demonstrates great insight by following the flow of the article despite this limitation.

I will correct the spelling errors in line 138 and 336.

Overall the review has brought new insights and important critical remarks for further research within this area.

Thank you very much.   

Reviewer 2 Report

The article deals with an important topic and sets out on a clear journey. While the theoretical section is of good quality, the section starting from par 3.1 is relatively weak and needs to be more detailed. E.g. it is not at all clear what the principles under 3.1. mean (364-373). More detail is needed to test these against the theoretical underpinnings. The evaluations are not linked to the principles presented or to the main question in a substantial way and this needs to be improved as well.

In addition there is quite a bit of repetition in the article. This would need to be checked against the other comments.

Author Response

The reviewer makes valuable comments that could improve the article. I am grateful for pointing out that section 3.1 needs more detail. I will expand on the principles.

I will make a thorough reading of the article and identify unnecessary repetitions.

Thank you for providing valuable comments to improve the article.

Round 2

Reviewer 2 Report

No further comments